# Prediction of diabetic kidney disease risk using machine learning models: A population-based cohort study of Asian adults

**Charumathi Sabanayagam[1,2]\*, Feng He[1], Simon Nusinovici[1], Jialiang Li[3], Cynthia Lim[4], Gavin Tan[1,2], Ching Yu Cheng[1,2]**

[1]Singapore Eye Research Institute, Singapore, Singapore; [2]Ophthalmology and Visual Sciences Academic Clinical Program, Duke-NUS Medical School, Singapore, Singapore; [3]Department of Statistics and Data Science, National University of Singapore, Singapore, Singapore; [4]Department of Renal Medicine, Singapore General Hospital, Singapore, Singapore

\*For correspondence: charumathi.sabanayagam@seri.com.sg

**Competing interest:** The authors declare that no competing interests exist.

## Abstract

**Background:** Machine learning (ML) techniques improve disease prediction by identifying the most relevant features in multidimensional data. We compared the accuracy of ML algorithms for predicting incident diabetic kidney disease (DKD).

**Methods:** We utilized longitudinal data from 1365 Chinese, Malay, and Indian participants aged 40–80 y with diabetes but free of DKD who participated in the baseline and 6-year follow-up visit of the Singapore Epidemiology of Eye Diseases Study (2004–2017). Incident DKD (11.9%) was defined as an estimated glomerular filtration rate (eGFR) <60 mL/min/1.73 m$^2$ with at least 25% decrease in eGFR at follow-up from baseline. A total of 339 features, including participant characteristics, retinal imaging, and genetic and blood metabolites, were used as predictors. Performances of several ML models were compared to each other and to logistic regression (LR) model based on established features of DKD (age, sex, ethnicity, duration of diabetes, systolic blood pressure, HbA1c, and body mass index) using area under the receiver operating characteristic curve (AUC).

**Results:** ML model Elastic Net (EN) had the best AUC (95% CI) of 0.851 (0.847–0.856), which was 7.0% relatively higher than by LR 0.795 (0.790–0.801). Sensitivity and specificity of EN were 88.2 and 65.9% vs. 73.0 and 72.8% by LR. The top 15 predictors included age, ethnicity, antidiabetic medication, hypertension, diabetic retinopathy, systolic blood pressure, HbA1c, eGFR, and metabolites related to lipids, lipoproteins, fatty acids, and ketone bodies.

**Conclusions:** Our results showed that ML, together with feature selection, improves prediction accuracy of DKD risk in an asymptomatic stable population and identifies novel risk factors, including metabolites.

**Funding:** This study was supported by the Singapore Ministry of Health's National Medical Research Council, NMRC/OFLCG/MOH-001327-03 and NMRC/HCSAINV/MOH-001019-00. The funders had no role in study design, data collection and analysis, decision to publish, or preparation of the manuscript.

## Editor's evaluation

The authors wanted to see which patients with diabetes develop kidney disease and outcomes. They used clinical characteristics, eye pictures, genetic factors and blood levels of metabolites, and they found a combination of these factors predicted kidney disease in people with diabetes.

## Introduction

Diabetes affected an estimated 415 million people worldwide in 2015, and this number is expected to increase to 642 million by 2040, with the greatest increase expected in Asia, particularly in India and China (*Ogurtsova et al., 2017*). With the rising prevalence of diabetes and an aging population, the burden of diabetic kidney disease (DKD), a leading cause of end-stage kidney disease (ESKD), cardiovascular disease (CVD), and premature deaths, is also set to rise in parallel. Diabetes accounts for 30–50% of all chronic kidney disease (CKD) cases, affecting 285 million people worldwide (*Webster et al., 2017*). As CKD is asymptomatic till more than 50% of kidney function decline, early detection of individuals with diabetes who are at risk of developing DKD may facilitate prevention and appropriate intervention for DKD (*Hill et al., 2016*; *Hirst et al., 2020*). However, early identification of individuals at risk of developing CKD in type 2 diabetes is challenging (*Alicic et al., 2017*). Although most ESKD cases are due to diabetes, awareness of diabetes as a risk factor for CKD is markedly lower in several countries, which may constitute a barrier for early detection of CKD (*White et al., 2008*; *Hussain et al., 2019*; *Couser et al., 2011*). Moreover, in people with diabetes, adherence to annual screening for DKD with estimated glomerular filtration rate (eGFR) and urine albumin-creatinine ratio (UACR) remains a challenge (*Manski-Nankervis et al., 2018*). Therefore, there is an urgent need for the characterization of new biomarkers to identify individuals at risk of progressive eGFR decline and enable timely intervention for improving outcomes in DKD (*Alicic et al., 2017*).

Several risk prediction models have been developed in the past for predicting progression to ESKD, but studies predicting onset of CKD in diabetic populations are limited. These studies were focused on clinical populations utilizing data from clinical trials (*Dunkler et al., 2015*) or heterogeneous cohorts of patients with different CKD definitions (*Jiang et al., 2020*). Dunkler et al. showed that albuminuria and eGFR were the key predictors and addition of demographic, clinical, or laboratory variables did not improve predictive performance beyond 69% (*Dunkler et al., 2015*). Current CKD risk prediction models developed using traditional regression models (e.g., logistic regression [LR] or linear regression) perform well when there are only small or moderate numbers of variables or predictors but tend to overfit if there is a large number of variables. Machine learning (ML) methods using 'Big data,' or multidimensional data, may improve prediction as they have fewer restrictive statistical assumptions compared to traditional regression models that assume linear relationships between risk factors and the logit of the outcomes and absence of multicollinearity among explanatory variables (*Sundström and Schön, 2020*; *Doupe et al., 2019*; *Bi et al., 2019*).

Diabetes is a metabolic disorder, and the metabolic changes associated with diabetes can lead to glomerular hypertrophy, glomerulosclerosis, tubulointerstitial inflammation, and fibrosis (*Alicic et al., 2017*). Several blood metabolites have been shown to be associated with DKD (*Colhoun and Marcovecchio, 2018*). Similarly, genetic abnormalities in diabetes have also been linked to an increased risk of DKD (*Cole and Florez, 2020*). We, and several others, have previously shown that retinal microvascular changes, including retinopathy, vessel narrowing, or dilation, and vessel tortuosity, were associated with CKD (*Yau et al., 2011*; *Yip et al., 2017*). Integrating high-dimensional data from multiple domains, including patient characteristics, clinical and 'Omics' data, has the potential to aid in risk stratification, prediction of future risk, and provide insights into the pathogenesis (*Eddy et al., 2020*). These features may contribute to prediction in very complicated ways, and they may not fully satisfy the requirement for a simple linear logistic model. It is thus more appropriate to consider ML approaches for a comprehensive study.

In the current study, we aimed to evaluate the performance of a set of the most common ML models, including traditional LR, for predicting the 6-year risk of DKD and identifying important predictors of DKD in a large population-based cohort study in Singapore with multidimensional data, including imaging, metabolites, and genetic biomarkers.

## Methods

### Study population

Data for this study was derived from the Singapore Epidemiology of Eye Diseases (SEED) study, a population-based prospective study of eye diseases in 10,033 Asian adults aged 40–80 y in Singapore. The follow-up study was conducted after a median duration of 6.08 y (interquartile range: [5.56, 6.79]), with 6762 participants. The detailed methodology of SEED has been published elsewhere (*Majithia et al., 2021*). Briefly, the name list of adults residing in the southwestern part of Singapore was provided by the Ministry of Home Affairs, and then an age-stratified random sampling procedure was conducted. A total of 3280 Malays (2004–2007) (*Foong et al., 2007*), 3400 Indians (2007–2009), and 3353 Chinese (2009–2011) (*Lavanya et al., 2009*) participated in the baseline study with response rates of 78.7, 75.6 and 72.8%, respectively. As all three studies followed the same methodology and were conducted in the same study clinic, we combined the three populations for the present study. For the current analysis, we included only those with diabetes, defined as random glucose ≥11.1 mmol/L, HbA1c ≥ 6.5% (48 mmol/mol), self-reported antidiabetic medication use, or having been diagnosed with diabetes by a physician. Of the 6762 participants who attended both baseline and follow-up visit, after excluding those without diabetes (n = 5307), prevalent CKD (n = 315), missing information on eGFR (n = 90), the final sample size for analysis was 1365 (47.5% Indians, 27.8% Malays, and 24.7% Chinese). The sample size available for each dataset after removing participants missing >10% data was between 976 and 1364 (*Supplementary file 1-Table 1a*). SEED was conducted in accordance with the Declaration of Helsinki and was approved by the SingHealth Centralised Institutional Review Board (2018/2717, 2018/2921, 2012/487/A, 2015/2279, 2018/2006, 2018/2594, 2018/2570). Informed consent was obtained from all participants.

### Assessment of DKD

Incident DKD was defined as an eGFR <60 mL/min/1.73 $m^2$ with at least a 25% decrease in eGFR at follow-up in participants who had eGFR ≥ 60 mL/min/1.73 $m^2$ at baseline. Combining change in eGFR category together with a minimal percent change ensures that small changes in eGFR, for example, from 61 to 59 mL/min/1.73 $m^2$, are not misinterpreted as incident CKD as the eGFR is < 60 mL/min/1.73 $m^2$ (*Yip et al., 2017*; *Stevens et al., 2013*). The reduction in eGFR at follow-up was calculated as a percentage of the baseline eGFR as (eGFR at baseline – eGFR at follow-up)/eGFR at baseline * 100%. GFR was estimated from plasma creatinine using the Chronic Kidney Disease Epidemiology Collaboration (CKD-EPI) equation (*Levey et al., 2009*). Blood creatinine was measured by the Jaffe method on the Beckman DXC800 analyzer calibrated to the Isotope Dilution Mass Spectrometry (IDMS) method using the National Institute of Standards and Technology (NIST) Reference material.

### Variables for prediction

We evaluated 339 features such as demographic, lifestyle, socioeconomic, physical, laboratory, retinal imaging, genetic and blood metabolomics profile. The entire list of variables is presented in *Supplementary file 1-Table 1b*. We organized the variables into five different domains: traditional risk factors, extended risk factors, imaging parameters, genetic parameters, and blood metabolites. For ML analysis, based on different combinations of the five domains, we tested six models (A to F): A = traditional risk factors; B = A + extended risk factors; C = B + imaging parameters; D = B + genetic parameters; E = B + blood metabolites; F = B + imaging parameters + blood metabolites + genetic parameters.

### Traditional risk factors (n = 7)

Age, sex, ethnicity (Chinese, Malays, and Indians), body mass index (BMI, kg/$m^2$), systolic blood pressure (BP, mm Hg), duration of diabetes (years), and HbA1c% were included as traditional risk factors.

### Extended risk factors (n = 22)

Participant information was collected using an interviewer-administered questionnaire (demographic, socioeconomic, lifestyle factors, and personal history of diseases and medication use), physical examination (height, weight, BP), and laboratory examination (blood glucose, creatinine, lipid profile) (*Majithia et al., 2021*). Marital status, educational level (primary/below [≤6 y], secondary/

above [7 y and above of education]), monthly income, smoking status (current smokers vs. former and nonsmokers), alcohol consumption (ever vs. nondrinkers), history of CVD (self-reported history of myocardial infarction, stroke or angina), hypertension status, diastolic BP, pulse pressure, blood glucose, total, high-density lipoprotein (HDL) and low-density lipoprotein (LDL) cholesterol levels, antidiabetic including oral hypoglycemia drugs and insulin, antihypertensive, and anticholesterol medication use were included as part of extended risk factors.

## Blood metabolites (n = 223)

We quantified 228 metabolic measures from stored serum/plasma samples at baseline using a high-throughput NMR metabolomics platform (Nightingale Health, Helsinki, Finland). The metabolites included routine lipids, lipoprotein subclasses with lipid concentrations within 14 subclasses, fatty acids, amino acids, ketone bodies, and glycolysis-related metabolites. The 14 lipoprotein subclasses include six subclasses of VLDL (extremely large, very large, large, medium, small, very small), IDL, three subclasses of LDL (large, medium, small), and four subclasses of HDL (very large, large, medium, small). Lipid concentration within each lipoprotein particle included triacylglycerol, total cholesterol, non-esterified cholesterol and cholesteryl ester levels, and phospholipid concentrations (*Quek et al., 2021*). Of the 228 metabolites, pyruvate, glycerol, and glycine were not available in Malays. In addition, creatinine and glucose were measured as part of the blood biochemistry. After excluding these five metabolites, 223 were included under the metabolites dataset.

## Genetic parameters (n = 76)

We included 76 type 2 diabetes-associated single-nucleotide polymorphisms (SNPs) identified in the largest meta-analysis of type 2 diabetes genome-wide association studies by the DIAbetes Genetics Replication and Meta-analysis consortium (*Chong et al., 2017*).

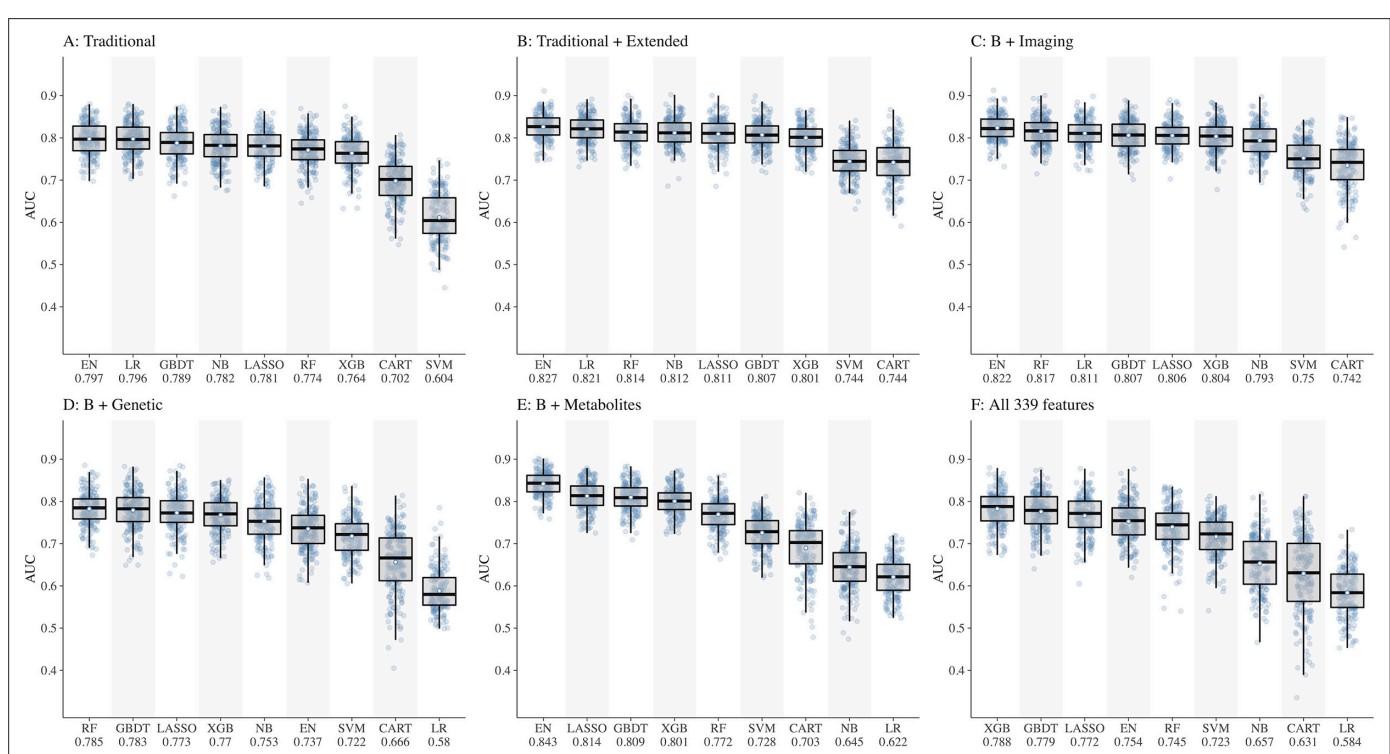

**Figure 1.** Comparison of nine machine learning models for diabetic kidney disease (DKD) incidence prediction using different sets of features (Panel A-F). Abbreviations: CART, classification and regression tree; EN, elastic net; GBDT, gradient boosting decision tree; LASSO, least absolute shrinkage and selection operator; LR, logistic regression; NB, naïve Bayes; RF, random forest; SVM, support vector machine; XGB, extreme gradient boosting.

### Imaging parameters (n = 11)

Using a semi-automated computer program (Singapore I Vessel Assessment, SIVA), we quantified retinal imaging parameters from digital retinal photographs. The parameters included retinal arteriolar and venular diameters, vessel tortuosity, branching angle, fractal dimension, etc. (*Yip et al., 2017*). Diabetic retinopathy (DR) was assessed by trained graders using a standard protocol (*Sabanayagam et al., 2019*).

### Machine learning algorithms

We tested nine different ML algorithms, including LR, least absolute shrinkage and selection operator (LASSO), elastic net (EN), classification and regression tree (CART), random forest (RF), gradient boosting decision tree (GBDT), extreme gradient boosting (XGB), support vector machine (SVM), and naïve Bayes (NB) (*Hastie et al., 2009*).

### Model development

We split the study samples randomly into training (80%) and test sets (20%) of equal CKD case rate by stratified sampling, with 40 random repeats of 5-fold cross-validation to evaluate the model performance. Predictive accuracy was assessed using metrics such as area under the receiver operating characteristic curve (AUC) with 95% confidence interval (CI) , sensitivity and specificity calculated at the optimal cut-point (determined by Youden's index). In preliminary analyses, testing different combinations of features (*Figure 1A–F*), performance of all ML models was below 0.80 in dataset D including genetic features (best AUC = 0.785 by RF) and dataset F including all 339 features (best AUC = 0.788 by XGB). Hence, we dropped these two datasets (D and F) from further analyses. The performance of all ML models based on AUC (IQR) in datasets 1A–F is shown in *Supplementary file 1-Table 1c* and based on sensitivity and specificity is shown in *Supplementary file 1-Table 1d*.

Of the ML models, performances of CART, SVM, and NB were lower compared to other models, hence these models were also dropped. Consequently, ML models EN, GBDT, LASSO, XGB, and RF were considered for subsequent analyses using datasets A, B, C, and E including 252 features.

### Feature selection

All algorithms included in the current study can perform feature selection but using different selection criteria. In LR, stepwise selection according to the Akaike information criterion (AIC) is widely used but it lacks stability. LASSO is an extension of LR with L1 regularization to drop the less important variables. EN is like LASSO but with a milder regularization, resulting in a larger number of retained variables. In order to select only the most predictive features, we recursively apply EN until the retained variable subset is optimized, that is, recursive feature selection (RFE). In RF, GBDT, and XGB, the most predictive variables were identified based on their relative importance to model performance. Feature selection was also performed according to their selection frequency during repeated cross-validation. We identified the top 15 predictors by each of the best-performing ML models, then compared the performance of the ML models based on the top variables with that of LR based on seven traditional risk factors (age, sex, ethnicity, BMI, HbA1c, duration of diabetes, and systolic BP) in another 40 random repeats of 5-fold cross-validation. Subgroup analyses were conducted for the three ethnic groups separately.

### Statistical analyses

We compared the baseline characteristics of participants with diabetes by incident DKD status using $\chi^2$ test or Mann–Whitney $U$ test as appropriate for the variable and compared the socioeconomic status by ethnicity using $\chi^2$ test and Kruskal–Wallis test as appropriate for the variable. Statistical significance was defined as a p-value<0.05. Subgroup numbers such as DR status may not add up due to the presence of missing data. For modeling, we used mean values/modes for missing value imputation as appropriate for each variable because the missing proportions were all below 10%. Improvement in prediction accuracy by ML over the traditional risk factor model was calculated as (ML AUC – traditional model AUC)/traditional model AUC * 100%. All analyses were conducted using R software version 4.0.2. To assess whether the features selected by ML models were meaningful, we visualized the association of top 15 variables with incident DKD in forest plots or a variable importance plot as appropriate for the algorithm.

**Table 1.** Baseline characteristics of SEED diabetic participants by incident DKD status.

| Characteristics | No DKD (n = 1203) | DKD (n = 162) | p-value | Overall (n = 1365) |
|---|---|---|---|---|
| Age (years) | 57.95 (8.78) | 64.63 (7.98) | <0.001 | 58.74 (8.95) |
| Sex, female | 580 (48.2) | 87 (53.7) | 0.219 | 667 (48.9) |
| Ethnicity | | | <0.001 | |
| Indians (ref) | 599 (49.8) | 49 (30.2) | | 648 (47.5) |
| Malays | 310 (25.8) | 70 (43.2) | | 380 (27.8) |
| Chinese | 294 (24.4) | 43 (26.5) | | 337 (24.7) |
| Primary/below education (%) | 706 (58.7) | 121 (74.7) | <0.001 | 827 (60.6) |
| Current smoker (%) | 173 (14.4) | 16 (9.9) | 0.15 | 189 (13.9) |
| Alcohol consumption (%) | 111 (9.2) | 11 (6.8) | 0.389 | 122 (9.0) |
| Hypertension (%) | 845 (70.4) | 155 (95.7) | <0.001 | 1000 (73.4) |
| Diabetic retinopathy (%) | 228 (19.2) | 56 (35.4) | <0.001 | 284 (21.1) |
| Cardiovascular disease (%) | 153 (12.7) | 32 (19.8) | 0.02 | 185 (13.6) |
| Duration of diabetes (years) | 2.68 [0.00, 8.56] | 6.08 [1.44, 11.63] | <0.001 | 3.20 [0.00, 9.37] |
| Antidiabetic medication (%) | 681 (56.6) | 122 (75.3) | <0.001 | 803 (58.8) |
| Insulin use (%) | 39 (3.3) | 11 (7.1) | 0.036 | 50 (3.8) |
| Body mass index (kg/m$^2$) | 26.96 (4.62) | 27.05 (4.36) | 0.764 | 26.97 (4.59) |
| Systolic blood pressure (mm Hg) | 139.42 (18.95) | 155.24 (20.01) | <0.001 | 141.29 (19.74) |
| Diastolic blood pressure (mm Hg) | 78.25 (9.74) | 79.14 (10.70) | 0.278 | 78.35 (9.85) |
| Random blood glucose (mmol/L) | 9.53 (4.26) | 10.44 (5.01) | 0.052 | 9.64 (4.36) |
| HbA1c (%) | 7.61 (1.58) | 8.04 (1.83) | 0.003 | 7.66 (1.62) |
| Blood total cholesterol (mmol/L) | 5.14 (1.14) | 4.98 (1.15) | 0.124 | 5.12 (1.15) |
| Blood HDL cholesterol (mmol/L) | 1.12 (0.31) | 1.16 (0.35) | 0.178 | 1.12 (0.32) |
| eGFR (mL/min/1.73 m$^2$) | 89.98 (14.34) | 79.40 (11.69) | <0.001 | 88.72 (14.46) |

Values for categorical variables are presented as number (percentages); values for continuous variables are given as mean (SD) or median [IQR]. p-values are given by $\chi^2$ test or Mann–Whitney $U$ test as appropriate for the variable.

DKD, diabetic kidney disease; HDL, high-density lipoprotein cholesterol; IQR, interquartile range; SD, standard deviation; SEED, Singapore Epidemiology of Eye Diseases.

## Results

The 6-year incidence of DKD was 11.9% in the study population. The incidence of DKD was the highest in Malays (18.4%), followed by Chinese (12.8%). Although Indians represent nearly half of the total diabetic population (648 of the 1365 diabetic participants, 47.5%), DKD incidence was the lowest in Indians (7.6%).

As shown in *Table 1*, compared to those without incident DKD, those with were significantly older, more likely to be Malays, primary/below educated, had higher prevalence of hypertension, DR, CVD, antidiabetic medication use; had longer duration of diabetes, and higher levels of systolic BP and HbA1c%.

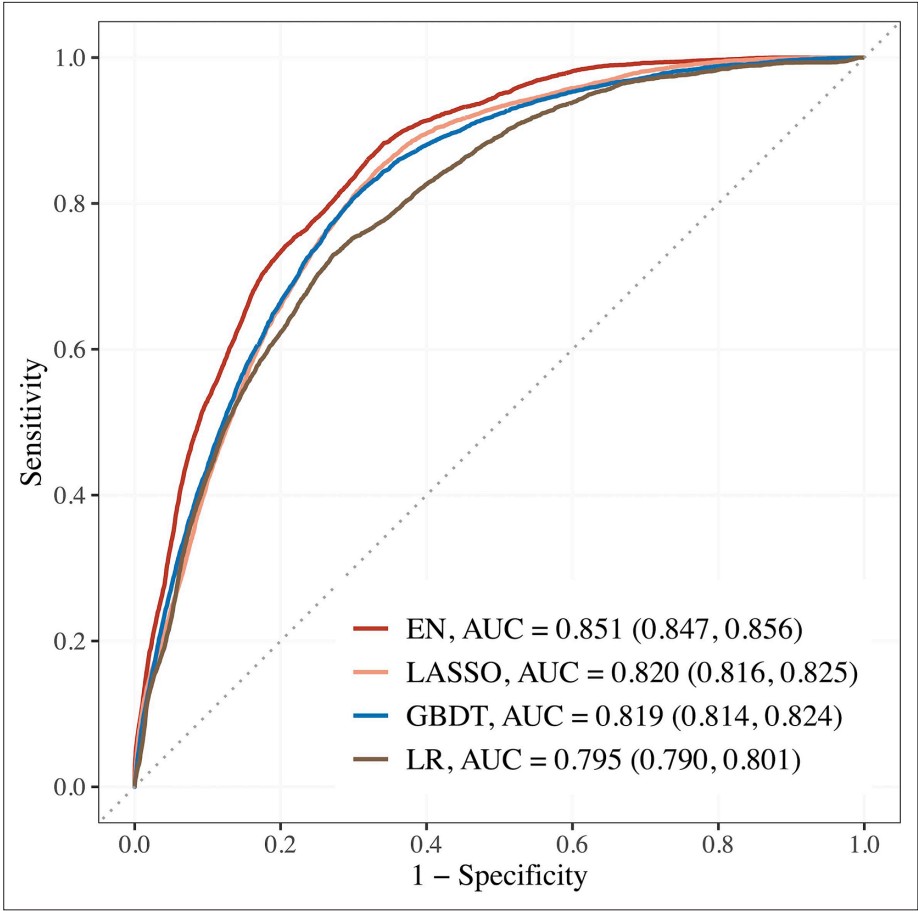

**Figure 2.** Performance of the top 3 machine learning (ML) models based on selected variables in dataset E (risk factors + blood metabolites) compared to LR using seven established features. Abbreviations: EN, Elastic net; GBDT, gradient boosting decision tree; LASSO; least absolute shrinkage and selection operator; LR, logistic regression.

## Performance of LR using traditional risk factors (reference) and other domain features

The LR using the seven traditional risk factors (age, sex, ethnicity, BMI, HbA1c, duration of diabetes, and systolic BP) had an AUC of 0.796. The performance of LR improved to 0.821 using the traditional + extended risk factors. With additional features, the performance of LR dropped significantly (AUC of 0.622 in E and 0.811 in C).

## Performance of ML models using multidimensional data

Using datasets, A, B, C, and E, the performances of the five ML models (*Figure 1A–C and E*) were as follows:

1. EN ranked first in performance in all four datasets with AUCs ranging from 0.797 in A to 0.843 in E
2. LASSO ranged from 0.781 in A to 0.814 in E
3. GBDT ranged from 0.789 in A to 0.809 in E
4. RF ranged from 0.772 in E to 0.817 in C
5. XGB ranged from 0.764 in A to 0.804 in C

*Figure 2* shows the AUCs of the top 3 performing models. Using the top 15 predictors generated by feature selection, the performance of EN improved further with an AUC (95% CI) of 0.851 (0.847–0.856), sensitivity and specificity of 88.2 and 65.9% compared to LR using seven established features with AUC of 0.795 (0.790–0.801), and sensitivity and specificity of 73.0 and 72.8%.

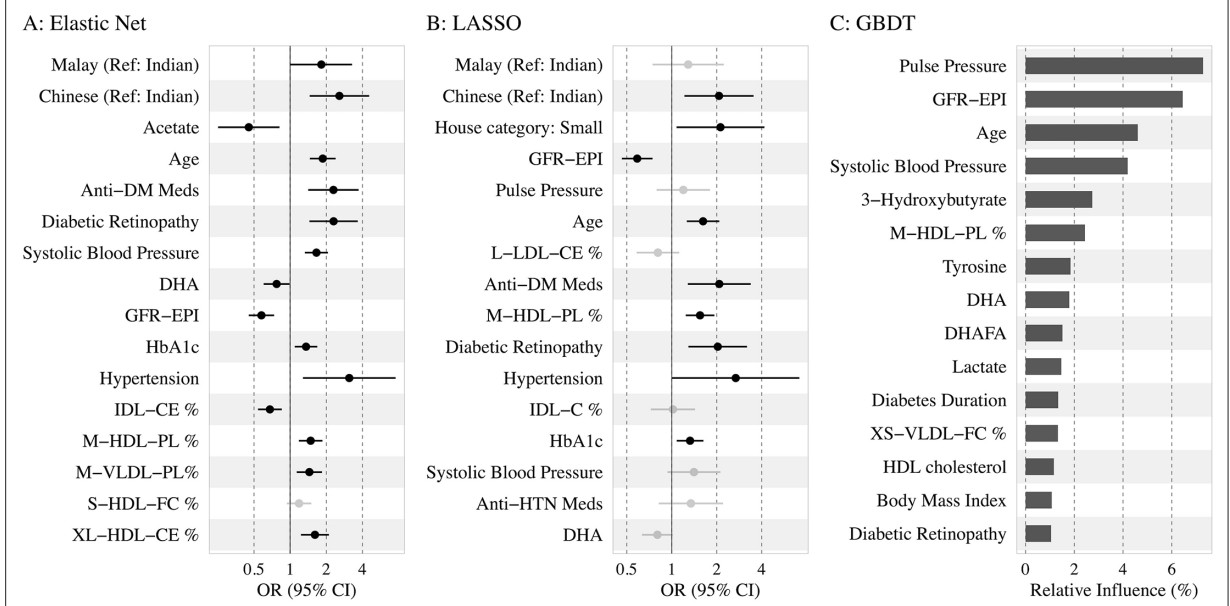

**Figure 3.** Association of the top 15 machine learning (ML)-selected predictors with incident diabetic kidney disease (DKD). Abbreviations: LASSO, least absolute shrinkage and selection operator; GBDT, Gradient boosting decision tree. Variables: anti-HTN Meds, anti hypertensive medications; DM, diabetes mellitus; GFR-EPI, glomerular filtration rate estimated using CKD-EPI equation; HDL, high-density lipoprotein. Metabolites: DHA: 22:6, docosahexaenoic acid; DHAFA, Ratio of 22:6 docosahexaenoic acid to total fatty acids; IDL-CE%, Cholesterol esters to total lipids ratio in IDL; M-HDL-PL%, Phospholipids to total lipids ratio in medium HDL; M-VLDL-PL%, Phospholipids to total lipids ratio in medium VLDL; S-HDL-FC% Free cholesterol to total lipids ratio in small HDL; XL-HDL-CE%, Cholesterol esters to total lipids ratio in very large HDL; L-LDL-CE%, Cholesterol esters to total lipids ratio in large LDL; M-HDL-PL%, Phospholipids to total lipids ratio in medium HDL; IDL-C% Total cholesterol to total lipids ratio in IDL; M-HDL-PL% Phospholipids to total lipids ratio in medium HDL.

The corresponding estimates for LASSO were 0.820 (0.816–0.825), 84.4% and 67.0%; 0.819 (0.814–0.824), 80.6% and 70.1% for GBDT. The AUCs of EN, LASSO, and GBDT were 7.0, 3.1, and 3.0% relatively higher than that of LR.

## Top 15 predictors

*Figure 3* shows the top 15 predictors visualized using forest plots for EN and LASSO and a variable importance plot for GBDT. We found low collinearity in EN-selected features (Spearman's correlation coefficients: –0.49 to 0.43), while LASSO and GBDT selected some variables of higher correlation (e.g., systolic BP and pulse pressure).

Among the traditional and extended risk factors, all three models chose age, SBP, DR, and lower levels of eGFR as the top 15 predictors. In addition, antidiabetic medication use, HbA1c, hypertension, and ethnicity (Malays and Chinese compared to Indians) were chosen as risk factors by EN and LASSO; antihypertensive medication and low housing type by LASSO; and duration of diabetes, BMI, and HDL cholesterol by GBDT. Among the metabolites, phospholipids to total lipids ratio in MHDL and DHA were selected by all three models. Free cholesterol to total lipids ratio in small HDL/XSVLDL, and cholesterol esters to total lipids ratio in IDL/LLDL/XLHDL were also found to be of high frequency. Additionally, higher levels of acetate were shown to be protective by LR based on EN-selected variables, while tyrosine and lactate were identified as important factors by GBDT. Source data for the forest plots are shown in *Supplementary file 1-Table 1e*. Using the same approach, we also identified the top variables by each ethnicity in *Supplementary file 1-Table 1f*, and found baseline eGFR linked to incident DKD in all three subgroups, whereas acetate, SBP, antidiabetic medication use, and housing type were important in two ethnic subgroups.

## Discussion

The results of the current study suggest that prediction using ML models with selected features provided improved prediction compared to LR model based on seven established features in this

**Table 2.** Machine learning model for predicting incident CKD in literature.

| Author, journal | Study cohort, country | Study population Follow-up | CKD definition and incidence | Number of predictors | ML performance |
|---|---|---|---|---|---|
| *Ravizza et al., 2019*, *Nature Medicine* | EHR data from the IBM Explorys and INPC datasets, the United States | Development cohort (IBM): >500,000 adults with diabetes. Validation (INPC) = 82,912 adults with T2DM; FU = 3 y. | ICD 9/10 codes | 300 features | Based on seven prioritized features, AUC by RF = 0.833 and the Roche/IBM supervised algorithm by LR = 0.827 |
| *Song et al., 2020*, *JMIR* | EHR data, the United States (2007–2017) | 14,039 adults with T2DM. FU = 1 y. | eGFR < 60 or UACR ≥30 mg/g; 34.1% | >3000 | GBM AUC = 0.83 |
| *Huang et al., 2020a*, *Diabetes* | KORA cohort, Germany | 1838 adults with prediabetes and T2DM. FU = 6.5 y. | eGFR < 60 or UACR ≥30 mg/g at FU; 10.9% | 125 mets + 14 clinical factors | SVM, RF, Ada Boost Best set: Mets-SM and PC + age, TC, FPG, eGFR, UACR, AUC = 0.857 Traditional LR using 14 variables, AUC = 0.809 |
| Sabanayagam et al., 2023, current study | SEED population data, Singapore | 1365 adults with diabetes. FU = 6 y. | eGFR < 60 + 25% decline in eGFR from baseline | 339 features | EN + RFE selected 15 features, AUC = 0.851 vs. 0.795 using seven features by traditional LR |

AUC, area under the receiver operating characteristic curve; CKD, chronic kidney disease; eGFR, estimated glomerular filtration rate; EHR, electronic health records; EN, Elastic Net; FPG, Fasting plasma glucose; FU, follow-up; GBM, Gradient Boosting Machine; ICD, International Classification of Diseases; INPC, Indiana Network for Patient Care; LR, logistic regression; ML, machine learning; RF, random forest; RFE, recursive feature selection; SEED, Singapore Epidemiology of Eye Diseases; SVM, support vector machine; T2DM, type 2 diabetes mellitus; TC, total cholesterol; UACR, urine albumin-creatinine ratio.

extensively phenotyped large-scale epidemiological study. The best performance was obtained by the EN model based on dataset E including risk factors and metabolites with an AUC of 0.851, which was 7.0% higher than that of LR using seven established risk factors. Sensitivity was also higher by EN (88.2 and 65.9%) compared to LR (73.0 and 72.8%). The top 15 predictors by EN using RFE identified several metabolites related to lipid concentration, lipoprotein subclasses, fatty acids, and ketone bodies as novel predictors besides confirming traditional predictors, including age, ethnicity, antidiabetic medication use, presence of hypertension, DR, higher levels of systolic BP, HbA1c, and lower levels of eGFR. Contrary to conventional risk factors, sex, BMI, and duration of diabetes did not come in the top 15-predictors.

Our results showed that ML models combined with feature selection improved the accuracy for predicting incident DKD in high-dimensional datasets. The AUC of MLs based on dataset E including metabolites (+risk factors) scored the highest, while the one based on dataset D including genetic features scored the lowest compared to other domain features. This finding suggests that modifiable risk factors and metabolites predict DKD risk better than genetic features. The predictive performance was the best by EN, followed by LASSO and GBDT. The top 15 predictors selected by LASSO and GBDT were largely consistent to that by EN.

Few previous studies have evaluated the performance of ML models for predicting the risk of incident DKD (*Table 2*). Ravizza et al. identified seven key features (age, BMI, eGFR, concentration of creatinine, glucose, albumin, and HbA1c%) by a data-driven feature selection strategy for predicting DKD using electronic health records (EHR) data from 417,912 people with diabetes retrieved from the IBM Explorys Database and developed a random forest model in 82,912 people with diabetes retrieved from the Indiana Network for Patient Care (INPC). The RF algorithm using seven prioritized key features achieved an AUC of 0.833 compared to 0.827 by LR (*Ravizza et al., 2019*).

Song et al. predicted 1-year risk of DKD based on electronic health records (EHR) data using gradient boosting machine (GBM) algorithm with an AUC of 83% (*Song et al., 2020*). As the median duration of development of DKD was ~10 y since the onset of diabetes, predicting 1-year risk may not be sufficient. Huang et al. predicted DKD risk in 1838 adults with diabetes and prediabetes who participated in the KORA study in Germany. The authors used ML models SVM, RF, and Ada Boost based on 14 clinical factors and 125 metabolites. The best achieved AUC was 0.857, which is similar to that of our model using EN (AUC = 0.851) (*Huang et al., 2020a*).

In the current study, we observed that when the features were limited to the traditional risk factors, the performance of LR was similar to that of the best ML model EN. However, when the number of features was large, LR's performance dropped significantly compared to the top-performing ML models, including EN, LASSO, and GBDT. While ML models are capable of addressing complex variable effects and nonlinearity issues, herein we found regularized regression models (EN and LASSO) outperforming other more sophisticated models. This suggests that the pathophysiological progression from diabetes to CKD may not be as nonlinear as previously assumed. At CKD stage 4, heterogeneity in terms of disease trajectory is primarily low, and nonadditive effects are likely to be negligible, which could explain the superior performance of EN and LASSO in our study. In a previous study based on the same dataset as the current study, Nusinovici et al. tested the performance of several ML models utilizing 20 risk factors alone and found that the performance of LR (AUC = 0.905) was similar to that of the best ML model, GBDT (AUC = 0.903), for predicting incident CKD in those with and without diabetes (*Nusinovici et al., 2020*). When a large number of features are present, more advanced ML methods may capture the complex functional dependency of the incident CKD outcome much better than the linear approach used in LR.

The top-performing ML models (EN, LASSO, GBDT) identified established risk factors for DKD, such as age, ethnicity, antidiabetic medication use, presence of hypertension, DR, higher levels of systolic BP, HbA1c, and lower levels of eGFR. Additionally, antihypertensive medication use and low housing type were identified by LASSO while BMI and duration of diabetes by GBDT. Increasing age, longer duration of diabetes, higher levels of HbA1c, systolic BP/hypertension are well-known risk factors of DKD. A meta-analysis conducted by Nelson et al., including 15 multinational cohorts with diabetes as part of the CKD Prognosis Consortium (*Nelson et al., 2019*), also identified older age, hypertension, lower eGFR, higher levels of BMI, HbA1c, and antidiabetic medication use as significant risk factors for incident CKD in those with diabetes. While black ethnicity was a risk factor for CKD in the meta-analysis, in our study, we found Chinese and Malay ethnicity to be at higher risk of developing incident DKD compared to Indian ethnicity. Compared to the Indian population, the Chinese population was generally older, while Malay participants had lower education levels and higher prevalence of smoking (*Supplementary file 1 -Table 1g*). Both Chinese and Malays had a higher prevalence of hypertension and lower levels of antidiabetic medication use, which may contribute to the ethnic difference in DKD incidence. Another reason for the lower risk of developing DKD in the Indian ethnicity could be that as a high-risk group for diabetes they may be more aware of the risk and comply with screening, medication, and other measures that can reduce their risk of developing DKD. Malay ethnicity has been identified to be a high-risk group for CKD by several studies conducted in Singapore (*Sabanayagam et al., 2010*; *Lim et al., 2021*; *National Registry of Diseases Office, 2020*). Surprisingly, sex was not identified to be a risk factor by any of the three ML models, which is consistent with the findings of the Ravizza et al. algorithm based on data-driven feature selection that did not pick up sex as one of the priority features (*Ravizza et al., 2019*).

In the current study, several new predictors from the metabolites domain were identified. Specifically, lipid metabolites, including phospholipids in HDL and VLDL subclasses, cholesterol esters, and free cholesterol in HDL subclasses, were found to be associated with an increased risk of DKD, while cholesterol esters in IDL were found to be protective against DKD. Furthermore, higher levels of DHA, acetate, and tyrosine also showed a protective association (odds ratios not shown). These findings are consistent with previous studies. For example, in the ADVANCE trial, higher tyrosine levels were associated with an increased risk of microvascular complications in diabetic participants. DHA, a n-3 polyunsaturated fatty acid (PUFA), has been shown to reduce renal inflammation and fibrosis and slow down the progression of CKD in animal models with type 2 diabetes (*Mathi Thumilan et al., 2016*). PUFA supplementation has also been shown to reduce hyperglycemia-induced pathogenic mechanisms by its anti-inflammatory and antioxidant properties, and improve renal function in diabetic nephropathy patients . Furthermore, higher levels of the short-chain fatty acid acetate have been shown to be inversely associated with diabetic nephropathy in type 2 diabetic patients (*Huang et al., 2020b*), and have beneficial effects in mice models with type 2 diabetes by reducing oxidative stress and inflammation.

The strengths of our study include a multiethnic Asian population with a long follow-up and the availability of a wealth of information. The use of RFE for dimension reduction and feature selection reduced overfitting of data. ML models identified the relative importance of one domain over the

other domains (like metabolite features in our study compared to genetic features) and the best predictors within one domain. However, our study results should be interpreted in light of a few limitations. First, our definition of DKD was based on the measurement of a single blood creatinine both at baseline and follow-up. This could have resulted in some misclassification, but the bias would be nondifferential and similar across both outcomes. Second, albuminuria, an important predictor of DKD, was not included as it was missing in a substantial number of participants. Third, external validation was not performed. Fourth, ML models are computationally intensive compared to traditional regression models.

In conclusion, in a population-based sample of multiethnic Asian adults, we found that EN with specific metabolites outperformed the current DKD risk prediction models using demographic and clinical variables. Our results provide evidence that combining metabolites and ML models could improve the prediction accuracy for DKD and that increasing the use of ML techniques may discover new risk factors for DKD. Further testing in external populations would support the validity of the model.

## Additional information

### Funding

| Funder | Grant reference number | Author |
|---|---|---|
| Singapore Ministry of Health's National Medical Research Council | NMRC/OFLCG/MOH-001327-03 | Gavin Tan |
| Singapore Ministry of Health's National Medical Research Council | NMRC/HCSAINV/MOH-001019-00 | Charumathi Sabanayagam |

The funders had no role in study design, data collection and interpretation, or the decision to submit the work for publication.

### Author contributions

Charumathi Sabanayagam, Conceptualization, Data curation, Supervision, Funding acquisition, Methodology, Writing – original draft, Writing – review and editing; Feng He, Data curation, Software, Formal analysis, Methodology; Simon Nusinovici, Cynthia Lim, Gavin Tan, Writing – review and editing; Jialiang Li, Supervision, Writing – review and editing; Ching Yu Cheng, Funding acquisition, Writing – review and editing

### Author ORCIDs

Charumathi Sabanayagam https://orcid.org/0000-0002-4042-4719
Feng He http://orcid.org/0000-0002-4717-2243

### Ethics

SEED was conducted in accordance with the Declaration of Helsinki and was approved by the SingHealth Centralised Institutional Review Board [2018/2717, 2018/ 2921, 2012/487/A, 2015/2279, 2018/2006, 2018/2594, 2018/2570]. Informed consent was obtained from all participants.

### Decision letter and Author response

Decision letter https://doi.org/10.7554/eLife.81878.sa1
Author response https://doi.org/10.7554/eLife.81878.sa2

## Additional files

### Supplementary files

• Supplementary file 1. Description of variables in each dataset and results from supplementary analyses (Table 1a- 1g). (Table 1a) Characteristics of the preprocessed datasets A–F. (Table 1b) List of variables used for DKD prediction. (Table 1c) Median AUC [IQR] performance of the ML models using datasets A–F. (Table 1d) Median SN%/SP% performance of the ML models using datasets A–F.

(Table 1e) Source data linked to *Figure 3*. (Table 1f) Top ML-selected predictors for incident DKD in each of the three ethnic groups by EN and RFE. (Table 1g) Baseline characteristics of SEED diabetic participants by ethnicity (n = 1365).

• MDAR checklist

### Data availability

The authors declare that the data supporting the findings of this study are available within the article and its supplementary materials. The individual participant-level data cannot be shared publicly due to privacy and confidentiality concerns. However, de-identified data may be available upon request with a project proposal to qualified individuals, subject to approval by the Singapore Eye Research Institute and access will require a data sharing agreement. Interested researchers can send data access requests to Prof. Ching-Yu Cheng at the Singapore Eye Research Institute using the following email address: cheng.ching.yu@seri.com.sg. Processed version of the datasets are provided in Supplementary file 1-Table 1a-1g.

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
