## [Editor Report]

The authors wanted to see which patients with diabetes develop kidney disease and outcomes. They used clinical characteristics, eye pictures, genetic factors and blood levels of metabolites, and they found a combination of these factors predicted kidney disease in people with diabetes.

---

## [Decision Letter]

**Decision letter after peer review:**

Thank you for submitting your article "Prediction of diabetic kidney disease risk using machine learning models: a population-based cohort study of Asian adults" for consideration by *eLife*. Your article has been reviewed by 3 peer reviewers, and the evaluation has been overseen by a Reviewing Editor and Martin Pollak as the Senior Editor. The reviewers have opted to remain anonymous.

Essential revisions:

1) Statistical significance versus clinical significance:

The authors seem to use recursive feature elimination to come up with a set of top features for each Ml algorithm and select features from a varied feature set. However, the authors may need to pay attention to what the features (that come up as significant) are trying to allude to? for e.g. the authors seem to have dropped the datasets with features that contain the genetic and imaging parameters: D= B+ Genetic parameters and F= B+ Imaging parameters+ Blood metabolites+ Genetic parameters. They provide reasons for the low performance of the ML models for dropping the features but do not elaborate on whether they investigated the reasons for the drop in performance.

2) The authors speak about the advantage of using ML approaches to overcome shortcomings of traditional assumptions from linear models, however, in the consideration of their covariates they might also want to understand the clinical association between some of their selected features. for e.g. BMI, HbA1c, duration of diabetes, and systolic BP may somehow not be entirely independent of each other (especially in the context of influencing one another and driving diabetes) and multi-collinearity may need to be looked into.

3) One of the biggest limitations of the study is its longitudinal nature with 6 yr timeframe for the development of DKD. That is a long timeframe to be able to discount factors other than those mentioned that could have affected the development of kidney disease. For example, patients with diabetes are also at risk for heart diseases, infections, and other hospital admissions, all of which can affect the development of kidney disease over that timeframe and haven't been controlled for in the dataset. Without controlling for these factors, the results have the risk of being hugely biased.

4) How were patients who died within that timeframe treated? Were they classified as DKD or censored? A competing risk methodology (with or without ML) might be better suited for this question or at least merit a sensitivity analysis with it.

5) The definition of DKD solely relies on a decrease in eGFR. Though understandable, it has the potential to be highly flawed, especially with factors discussed in #1. Adding measures to proteinuria/albuminuria, if available, would greatly add to its value.

6) The authors define incident DKD as eGFR < 60. Albuminuria is generally the earliest sign of DKD – the authors say this is because of missing data. This omission may underestimate the incidence of CKD in this study. Screening for DKD with annual UACR should be addressed as well in the introduction, as the authors say that early detection is challenging.

7) Ethnicity should not be included as a "traditional risk factor" as it is not a biological variable, but rather a social construct. How was ethnicity determined in the SEED study?

8) What was the rationale for the hundreds of metabolites that were chosen?

9) We suggest performing subgroup analysis on each of the 3 ethnic groups (Chinese, Malay, and Indian) to look for differences that may be explained by other variables as all "Asians" are not the same.

*Reviewer #1 (Recommendations for the authors):*

I would recommend the authors look at their wealth of features and their data and perform more association analysis and try and explain their feature selections instead of just depending on the outcome from the RFE.

I would also encourage the authors to look at the development of the model in association with a clinician and a biostatistician so as to understand the outcome of the model in the context of the disease and explain what the model outcomes are telling them about the progression of the disease.

The authors should also try and address the issue of bias that can creep into the model due to the features selected. is the model giving you the 6 yr risk of developing CKD and stating that Malays and Chinese populations are at higher risk for CKD from diabetes or does the model allude to some sort of socioeconomic factors playing a role in not getting medically examined regularly enough to miss the progression of diabetes to CKD?

*Reviewer #2 (Recommendations for the authors):*

It is an interesting study using various supervised machine learning methods to identify the risk and risk factors for the development of diabetic kidney disease. The study is overall well done. Few comments –

1. One of the biggest limitations of the study is its longitudinal nature with 6 yr timeframe for the development of DKD. That is a long timeframe to be able to discount factors other than those mentioned that could have affected the development of kidney disease. For example, patients with diabetes are also at risk for heart diseases, infections, and other hospital admissions, all of which can affect the development of kidney disease over that timeframe and haven't been controlled for in the dataset. Without controlling for these factors, the results have the risk of being hugely biased.

2. How were patients who died within that timeframe treated? Were they classified as DKD or censored? A competing risk methodology (with or without ML) might be better suited for this question or at least merit a sensitivity analysis with it.

3. The definition of DKD solely relies on a decrease in eGFR. Though understandable, it has the potential to be highly flawed especially with factors as discussed in #1. Adding measures to proteinuria/albuminuria, if available, would greatly add to its value.

4. Another 2 questions that I wasn't able to find answers to were – the average length of follow-up and how much was loss to follow-up.

*Reviewer #3 (Recommendations for the authors):*

1. The authors define incident DKD as eGFR < 60. Albuminuria is generally the earliest sign of DKD – the authors say this is because of missing data. This omission may underestimate the incidence of CKD in this study. Screening for DKD with annual UACR should be addressed as well in the introduction, as the authors say that early detection is challenging.

2. Ethnicity should not be included as a "traditional risk factor" as it is not a biological variable, but rather a social construct. How was ethnicity determined in the SEED study?

3. What was the rationale for the hundreds of metabolites that were chosen?

4. Suggest performing subgroup analysis on each of the 3 ethnic groups (Chinese, Malay, and Indian) to look for differences that may be explained by other variables as all “Asians” are not the same.

5. There are several mentions in the discussion of ethnicity being a risk factor for CKD or diabetes – ethnicity is not the risk factor.

6. This sentence is problematic on page 16: "One reason for the Indian ethnicity to be at lower risk of developing DKD could be Indian ethnicity being a high-risk group for diabetes, they may be well aware of the risk, and comply with screening, medication, etc. that could reduce their risk of developing DKD."

7. Suggest expanding the discussion of metabolites: why were they chosen and what is the proposed mechanism that increases the risk of DKD?

8. What were the "anti-DM" meds? What percentage were insulin-dependent? This would suggest potentially higher DM severity.

9. Suggest discussing the “top 15” predictors in the 3 models and proposed mechanisms that are not traditional risk factors (e.g. 3-hydroxybutyrate in the GBDT model; acetate in the EN model).

[Editors’ note: further revisions were suggested prior to acceptance, as described below.]

Thank you for resubmitting your work entitled "Prediction of diabetic kidney disease risk using machine learning models: a population-based cohort study of Asian adults" for further consideration by *eLife*. Your revised article has been evaluated by Martin Pollak (Senior Editor) and a Reviewing Editor.

The manuscript has been improved but there are some remaining issues that need to be addressed, as outlined below:

*Reviewer #2 (Recommendations for the authors):*

The authors have answered most of the questions adequately. One major concern I still have is its longitudinal nature with 6 yr timeframe for the development of DKD. The authors have noted that as the AUC of the model is 0.85, baseline characteristics are good enough to predict DKD. If we look closely though the incidence of DKD was <12%. It is thus an unbalanced dataset and therefore AUC is likely an inflated value that needs to be interpreted with caution. I realize that ultimately this is the data that the authors have and it is still a valuable addition to the literature but this is an important limitation that needs to be acknowledged upfront.

---

## [Author Response]

Essential revisions:1) Statistical significance versus clinical significance:The authors seem to use recursive feature elimination to come up with a set of top features for each Ml algorithm and select features from a varied feature set. However, the authors may need to pay attention to what the features (that come up as significant) are trying to allude to? for e.g. the authors seem to have dropped the datasets with features that contain the genetic and imaging parameters: D= B+ Genetic parameters and F= B+ Imaging parameters+ Blood metabolites+ Genetic parameters. They provide reasons for the low performance of the ML models for dropping the features but do not elaborate on whether they investigated the reasons for the drop in performance.

Thank you for the question. We would like to clarify that our decision to drop genetic features and imaging parameters was based on both statistical and clinical significance.

Although not included in the main text, we analysed an additional set of features: B + Imaging parameters + Blood metabolites, getting median AUC scores of 0.840 by EN, 0.812 by LASSO, 0.806 by GBDT, 0.804 by XGB, 0.774 by RF, 0.727 by SVM, 0.688 by CART, 0.665 by NB, and 0.617 by LR. The results showed that imaging parameters did not provide additional predictive value beyond what was already captured by the metabolites (set E). Furthermore, adding genetic factors to set D and F always resulted in a decline in performance, indicating that genetic data carried more noises than signals.

From a clinical perspective, DKD is driven by multiple causes from different domains, including genetics, metabolites, environmental exposures, etc. While imaging parameters based on retinal fundus photography may provide insights into the microvasculature patterns in end-organs, they are indirectly related to kidney function. On the other hand, circulating metabolites are directly associated with kidney function since the kidney plays a critical role in metabolism. Genetic factors are also directly linked, but may not reflect the dynamic changes of bio-chemical reactions that may show early signs of DKD, and they do not always represent environmental exposures such as lifestyle and socio-economic status. Therefore, we believe that metabolites are more clinically relevant for predicting DKD in our population. Moreover, given the correlation between metabolites, genetics, and retinal micro-vasculature, after dropping certain features due to their lower predictive value or higher noise, some of their information may still be available through the selected metabolites or extended risk factors (e.g., diabetic retinopathy). We hope this clarifies our rationale for selecting features for our models.

2) The authors speak about the advantage of using ML approaches to overcome shortcomings of traditional assumptions from linear models, however, in the consideration of their covariates they might also want to understand the clinical association between some of their selected features. for e.g. BMI, HbA1c, duration of diabetes, and systolic BP may somehow not be entirely independent of each other (especially in the context of influencing one another and driving diabetes) and multi-collinearity may need to be looked into.

Thank you for the comment. We have taken into account the possibility of multicollinearity among our selected features and have evaluated the correlation between them using Spearman's correlation coefficients. For our primary model based on the EN algorithm, we found low levels of multicollinearity among any pairs of continuous variables, with correlation coefficients ranging from -0.49 to 0.43. We further assessed the generalized variance inflation factor (GVIF) in a multivariable logistic regression setting, and found that GVIF^(1/(2*Df)) ranged from 1.1 to 1.3, indicating low levels of multicollinearity.

However, we did observe high levels of multicollinearity between certain variables in the LASSO and GBDT models, including L-LDL-CE% and IDL-C% (=0.85), systolic BP and pulse pressure (=0.85), and DHA and HAFA (=0.76). This is not unexpected as machine learning algorithms like LASSO and GBDT were designed to be robust even in the presence of multicollinearity^1-3^.

Overall, we agree that the potential for multicollinearity among our selected features is an important consideration. We have evaluated this possibility and believe that the primary model based on the EN-selected variables is valid given its low levels of multicollinearity among the variables. We have added in the manuscript, results that:

“We found low collinearity in EN-selected features (Spearman's correlation coefficients: -0.49 to 0.43), while LASSO and GBDT selected some variables of higher correlation (e.g., systolic BP and pulse pressure).”

3) One of the biggest limitations of the study is its longitudinal nature with 6 yr timeframe for the development of DKD. That is a long timeframe to be able to discount factors other than those mentioned that could have affected the development of kidney disease. For example, patients with diabetes are also at risk for heart diseases, infections, and other hospital admissions, all of which can affect the development of kidney disease over that timeframe and haven't been controlled for in the dataset. Without controlling for these factors, the results have the risk of being hugely biased.

Thank you for this comment. We agree that the development of DKD is associated with various factors including heart disease, infections, and hospital admissions over the 6-year timeframe, which may not be captured in our dataset. However, our aim was to evaluate the predictive performance of baseline information alone. Our model achieved an AUC of over 0.85, indicating that baseline information alone could predict DKD incidence well.

At baseline, we controlled for comorbidity features such as CVD history and hypertension status, medication use (anti-diabetic, anti-hypertensive, and anti-cholesterol), and lifestyles such as smoking and alcohol consumption, all of which could contribute to later development of DKD.

4) How were patients who died within that timeframe treated? Were they classified as DKD or censored? A competing risk methodology (with or without ML) might be better suited for this question or at least merit a sensitivity analysis with it.

Thank you for your comment.

Participants who died within the timeframe were treated as missing data and were excluded from the analysis. Although a competing risk model could be useful, it would require more detailed information about the timing and cause of events. However, in our study we do not have regular eGFR measurements to monitor the exact time of the event. Additionally, as only a few participants died during the 6-year follow-up period^4^, we believe that excluding them from the analysis would not have a significant impact on our results. However, we acknowledge that a sensitivity analysis with a competing risk model could be useful in future research.

5) The definition of DKD solely relies on a decrease in eGFR. Though understandable, it has the potential to be highly flawed, especially with factors discussed in #1. Adding measures to proteinuria/albuminuria, if available, would greatly add to its value.

Thank you for the suggestion to add measures of proteinuria/ albuminuria to our definition of DKD. We acknowledge that including these measures would provide valuable information, but due to missing urine samples in a large proportion of Malay participants, adding proteinuria/albuminuria as part of the outcome definition would result in the exclusion of these participants and potentially biased results. We have mentioned this as a limitation in the Discussion section.

Our definition of DKD incidence focused on the development of DKD stage G3/worse, which does not rely on proteinuria/ albuminuria status^5^. This is still a clinically relevant outcome, as patients in this stage require close monitoring and treatment, which bears higher healthcare costs compared to those in stage G2/below.

6) The authors define incident DKD as eGFR < 60. Albuminuria is generally the earliest sign of DKD – the authors say this is because of missing data. This omission may underestimate the incidence of CKD in this study. Screening for DKD with annual UACR should be addressed as well in the introduction, as the authors say that early detection is challenging.

Thank you for bringing this to our attention. We would like to clarify that our study was focused on predicting future incidence of clinically significant DKD (moderate and above DKD), defined as DKD stage G3 and above according to the international guideline^5^, with at least 25% decline in eGFR, in participants who may or may not have albuminuria as early signs of DKD at baseline. Based on this definition, we believe that the incidence of clinically significant DKD is unlikely to have been underestimated, as it does not depend on the availability of albuminuria data.

While we agree that albuminuria is a critical biomarker for the early signs of DKD, large intra-individual variability in UACR has been reported^6^. Given that we only had a single UACR record taken at baseline, its usefulness for prediction purposes may be limited in our study. Additionally, due to missing urine samples in a large proportion of Malay participants, addressing albuminuria as part of the study would result in the exclusion of these participants and potentially biased results.

Still, we have included a sentence in the introduction that “…in people with diabetes, adherence to annual screening for DKD with eGFR and urine albumin-creatinine ratio (UACR) remains a challenge…”

7) Ethnicity should not be included as a "traditional risk factor" as it is not a biological variable, but rather a social construct. How was ethnicity determined in the SEED study?

Thank you for this comment.

In SEED, ethnicity information was retrieved from the Singapore government database using Singapore national identification number unique to each individual. In Singapore, the three major ethnicities have distinct characteristics, such as lifestyles and health profiles. For example, Malays typically abstain from alcohol consumption due to religious reasons (Supplementary Table S7). As a result, ethnicity has been included as a risk factor in many population-based studies in Singapore. Our previous studies have demonstrated that Malays have a higher risk of kidney disease while Indians have a higher risk of diabetes^7^.

Furthermore, in non-Asian population-based studies, certain ethnicities/race have been shown to have a higher risk for kidney function decline. For example, in CKD-EPI equation^8^, black race was given a distinct coefficient in eGFR calculation to account for the added risk when compared to other races^8^.

8) What was the rationale for the hundreds of metabolites that were chosen?

Thank you for this question. As stated in the methodology section, all metabolites were quantified using the same high-throughput NMR metabolomics platform provided by Nightingale Health Ltd., Helsinki, Finland in 2017 (https://nightingalehealth.com). Technical details and epidemiological applications of the metabolic biomarker data have been reviewed in several publications^9,10^. Herein we included all metabolites available from the platform, because diabetes is a metabolic disorder associated with multiple pathways, and the kidney is also involved in many metabolic processes. Furthermore, including all the metabolites would allow us to evaluate the effectiveness of machine learning in selecting the most relevant features. In conclusion, we believe that the inclusion of a wide range of metabolites allowed us to explore new pathways and identify novel factors associated with incident DKD using machine learning, and our findings would provide insights into the potential role of metabolic dysregulation in the development of DKD in multi-ethnic populations.

9) We suggest performing subgroup analysis on each of the 3 ethnic groups (Chinese, Malay, and Indian) to look for differences that may be explained by other variables as all "Asians" are not the same.

Thank you for your suggestion.

We have performed subgroup analysis on each of the 3 ethnic groups, and have included the new results in Supplementary Table S6. We have also added some description in the main text result section that:

“Using the same approach, we also identified the top variables by each ethnicity in Supplementary Table S6, and found baseline eGFR linked to incident DKD in all 3 subgroups, whereas acetate, SBP, anti-diabetic medication use, and housing type were important in 2 ethnic subgroups.”

Reviewer #1 (Recommendations for the authors):I would recommend the authors look at their wealth of features and their data and perform more association analysis and try and explain their feature selections instead of just depending on the outcome from the RFE.

Thank you for your insightful comment. Our previous studies have shown the association between CKD/DKD and most of the features selected by machine learning, including age, sex, ethnicity^11-13^, socioeconomic status^14^, higher blood pressure^15^, glycemic control^16^, obesity^17^, and retinal imaging markers^18-20^. We will consider incorporating additional analyses in future studies to provide a more comprehensive understanding of the underlying associations between serum metabolites and DKD incidence.

I would also encourage the authors to look at the development of the model in association with a clinician and a biostatistician so as to understand the outcome of the model in the context of the disease and explain what the model outcomes are telling them about the progression of the disease.

Thank you for the suggestion. We agree that careful interpretation of machine learning results is crucial. It is worth noting that our study was conducted by an interdisciplinary team, involving researchers from the Department of Statistics and Data Science at the National University of Singapore (Prof Li Jialiang) and the Department of Renal Medicine at the Singapore General Hospital (Dr Cynthia Lim). Hence the expertise of both clinical and statistical fields was combined to ensure the development of a robust and clinically meaningful model.

The authors should also try and address the issue of bias that can creep into the model due to the features selected. is the model giving you the 6 yr risk of developing CKD and stating that Malays and Chinese populations are at higher risk for CKD from diabetes or does the model allude to some sort of socioeconomic factors playing a role in not getting medically examined regularly enough to miss the progression of diabetes to CKD?

Thank you for bringing this to our attention. Ethnicity has been established as a routine risk factor in many population-based studies in Singapore because the three major ethnicities are different in terms of genetic variants, lifestyles, health profiles, etc^7^.

In this study, we have also taken into account socio-economic status factors such as marital status, education level, income, and housing category in feature selection (supplementary Table S2). However, the results indicated that these factors were not as influential as the variable ethnicity in predicting the risk of DKD incidence. This suggests that the ethnic difference in DKD incidence cannot be fully attributed to socio-economic factors alone.

Still to investigate the ethnic differences contributed by socio-economic factors, we have performed additional analysis and have added our findings in the discussion that:

“Compared to the Indian population, the Chinese population was generally older, while Malay participants had lower education levels and higher prevalence of smoking (Supplementary Table S7). Both Chinese and Malays had a higher prevalence of hypertension and lower levels of anti-diabetic medication use, which may contribute to the ethnic difference in DKD incidence”.

Reviewer #2 (Recommendations for the authors):It is an interesting study using various supervised machine learning methods to identify the risk and risk factors for the development of diabetic kidney disease. The study is overall well done. Few comments –1. One of the biggest limitations of the study is its longitudinal nature with 6 yr timeframe for the development of DKD. That is a long timeframe to be able to discount factors other than those mentioned that could have affected the development of kidney disease. For example, patients with diabetes are also at risk for heart diseases, infections, and other hospital admissions, all of which can affect the development of kidney disease over that timeframe and haven't been controlled for in the dataset. Without controlling for these factors, the results have the risk of being hugely biased.

(Similar to Essential revisions 3) Thank you for this comment. We agree that the development of DKD is associated with various factors including heart disease, infections, and hospital admissions over the 6-year timeframe, which may not be captured in our dataset. However, our aim was to evaluate the predictive performance of baseline information alone. Our model achieved an AUC of over 0.85, indicating that baseline information alone could predict DKD incidence well.

At baseline, we controlled for comorbidity features such as CVD history and hypertension status, medication use (anti-diabetic, anti-hypertensive, and anti-cholesterol), and lifestyles such as smoking and alcohol consumption, all of which could contribute to later development of DKD.

2. How were patients who died within that timeframe treated? Were they classified as DKD or censored? A competing risk methodology (with or without ML) might be better suited for this question or at least merit a sensitivity analysis with it.

(Similar to Essential revisions 4) Thank you for your comment. Participants who died within the timeframe were treated as missing data and were excluded from the analysis. Although a competing risk model could be useful, it would require more detailed information about the timing and cause of events. However, in our study we do not have regular eGFR measurements to monitor the exact time of the event. Additionally, as only a few participants died during the 6-year follow-up period, we believe that excluding them from the analysis would not have a significant impact on our results. However, we acknowledge that a sensitivity analysis with a competing risk model could be useful in future research.

3. The definition of DKD solely relies on a decrease in eGFR. Though understandable, it has the potential to be highly flawed especially with factors as discussed in #1. Adding measures to proteinuria/albuminuria, if available, would greatly add to its value.

(Similar to Essential revisions 5) Thank you for the suggestion to add measures of proteinuria/ albuminuria to our definition of DKD. We acknowledge that including these measures would provide valuable information, but due to missing urine samples in a large proportion of Malay participants, adding proteinuria/albuminuria as part of the outcome definition would result in the exclusion of these participants and potentially biased results.

Our definition of DKD incidence focused on the development of DKD stage G3/worse, which does not rely on proteinuria/ albuminuria status^5^. This is still a clinically relevant outcome, as patients in this stage require close monitoring and treatment, which bears higher healthcare costs compared to those in stage G2/below.

4. Another 2 questions that I wasn't able to find answers to were – the average length of follow-up and how much was loss to follow-up.

Thank you for this comment. In methodology we have indicated that:

“The follow-up study was conducted after a median duration of 6.08 years (interquartile range: [5.56, 6.79]) with 6,762 participants.”

Reviewer #3 (Recommendations for the authors):1. The authors define incident DKD as eGFR < 60. Albuminuria is generally the earliest sign of DKD – the authors say this is because of missing data. This omission may underestimate the incidence of CKD in this study. Screening for DKD with annual UACR should be addressed as well in the introduction, as the authors say that early detection is challenging.

(Similar to Essential revisions 7) Thank you for bringing this to our attention. We would like to clarify that our study was focused on predicting future incidence of clinically significant DKD (moderate and above DKD), defined as DKD stage G3 and above according to the international guideline^5^, with at least 25% decline in eGFR, in participants who may or may not have albuminuria as early signs of DKD at baseline. Based on this definition, we believe that the incidence of clinically significant DKD is unlikely to have been underestimated, as it does not depend on the availability of albuminuria data.

While we agree that albuminuria is a critical biomarker for the early signs of DKD, large intra-individual variability in UACR has been reported^6^. Given that we only had a single UACR record taken at baseline, its usefulness for prediction purposes may be limited in our study. Additionally, due to missing urine samples in a large proportion of Malay participants, addressing albuminuria as part of the study would result in the exclusion of these participants and potentially biased results.

Still, we have included a sentence in the introduction that “… in people with diabetes, adherence to annual screening for DKD with eGFR and urine albumin-creatinine ratio (UACR) remains a challenge …”

2. Ethnicity should not be included as a "traditional risk factor" as it is not a biological variable, but rather a social construct. How was ethnicity determined in the SEED study?

(Similar to Essential revision 8) Thank you for this comment.

In SEED, ethnicity information was retrieved from the Singapore government database using Singapore national identification number unique to each individual. In Singapore, the three major ethnicities have distinct characteristics, such as lifestyles and health profiles. For example, Malays typically abstain from alcohol consumption due to religious reasons (Supplementary Table S7). As a result, ethnicity has been included as a risk factor in many population-based studies in Singapore. Our previous studies have demonstrated that Malays have a higher risk of kidney disease while Indians have a higher risk of diabetes^7^.

Furthermore, in non-Asian population-based studies, certain ethnicities/race have been shown to have a higher risk for kidney function decline. For example, in CKD-EPI equation^8^, black race was given a distinct coefficient in eGFR calculation to account for the added risk when compared to other races^8^.

3. What was the rationale for the hundreds of metabolites that were chosen?

(Similar to Essential revisions 9) Thank you for this question. As stated in the methodology section, all metabolites were quantified using the same high-throughput NMR metabolomics platform provided by Nightingale Health Ltd., Helsinki, Finland in 2017 (https://nightingalehealth.com). Technical details and epidemiological applications of the metabolic biomarker data have been reviewed in several publications^9,10^. Herein we included all metabolites available from the platform, because diabetes is metabolic disorder associated with multiple pathways, and the kidney is also involved in many metabolic processes. Furthermore, including all metabolites would allow us to evaluate the effectiveness of machine learning in selecting the most relevant features. In conclusion, we believe that the inclusion of a wide range of metabolites allowed us to explore new pathways and identify novel factors associated with incident DKD using machine learning, and our findings would provide insights into the potential role of metabolic dysregulation in the development of DKD in multi-ethnic populations.

4. Suggest performing subgroup analysis on each of the 3 ethnic groups (Chinese, Malay, and Indian) to look for differences that may be explained by other variables as all “Asians” are not the same.

(Similar to Essential revisions 10) Thank you for bringing this to our attention. We have performed subgroup analysis on each of the 3 ethnic groups, and have added in the main text result section that:

“Using the same approach, we also identified the top variables by each ethnicity in Supplementary Table S6, and found baseline eGFR linked to incident DKD in all 3 subgroups, whereas acetate, SBP, anti-diabetic medication use, and housing type were important in 2 ethnic subgroups.”

5. There are several mentions in the discussion of ethnicity being a risk factor for CKD or diabetes – ethnicity is not the risk factor.

(Similar to Essential revision 8) Thank you for this comment.

The three major ethnicities in Singapore have distinct characteristics, such as lifestyles and health profiles. Hence, ethnicity has been included as a routine risk factor in many population-based studies in Singapore^11,12^. Our previous studies have also demonstrated that Malays have a higher risk of kidney disease while Indians have a higher risk of diabetes^7^.

Furthermore, in non-Asian population-based studies, certain ethnicities/race to have been shown to have a higher risk for kidney function decline. For example, in CKD-EPI equation^8^, black race was given a distinct coefficient in eGFR calculation to account for the added risk when compared to other races^8^.

6. This sentence is problematic on page 16: "One reason for the Indian ethnicity to be at lower risk of developing DKD could be Indian ethnicity being a high-risk group for diabetes, they may be well aware of the risk, and comply with screening, medication, etc. that could reduce their risk of developing DKD."

Thank you for binging this to our attention. We have revised the sentence to:

“Another reason for the lower risk of developing DKD in the Indian ethnicity could be that as a high-risk group for diabetes, they may be more aware of the risk and comply with screening, medication, and other measures that can reduce their risk of developing DKD.”

7. Suggest expanding the discussion of metabolites: why were they chosen and what is the proposed mechanism that increases the risk of DKD?

Thank you for your suggestion. We agree that discussing the mechanism of metabolites in DKD development is an interesting point. However, it is important to note that while these metabolites were selected by machine learning, their relationship with DKD incidence may not be causal and further research is needed to fully understand their significance. As such, caution should be exercised in interpreting the results. Investigating the mechanism further would be valuable, and we will consider this point in future research.

8. What were the "anti-DM" meds? What percentage were insulin-dependent? This would suggest potentially higher DM severity.

Thank you. As mentioned in the methodology section, anti-diabetic medication included both oral hypoglycemic drugs and insulin. Among the 803 individuals who were taking anti-diabetic medication at baseline, only 50 were on insulin therapy, which represents approximately 6% of the anti-diabetic medication use, and approximately 3% of the entire study population (n=1365). While we acknowledge that insulin therapy is an important indicator of diabetes mellitus (DM) severity, we have included other variables such as the duration of diabetes and HbA1c in our model to reflect the severity of DM. We have added the insulin use data in supplementary Table S7.

9. Suggest discussing the “top 15” predictors in the 3 models and proposed mechanisms that are not traditional risk factors (e.g. 3-hydroxybutyrate in the GBDT model; acetate in the EN model).

(Similar to Reviewer #3 point 7) Thank you for your suggestion. We agree that discussing the mechanism of the selected metabolites in DKD development is an interesting point. However, it is important to note that while these metabolites were selected by machine learning, their relationship with DKD incidence may not be causal and further research is needed to fully understand their significance. As such, caution should be exercised in interpreting the results. Investigating the mechanism further would be valuable, and we will consider this point in future research.

References:

1. Hastie T, Qian J, Tay K. An Introduction to glmnet. *CRAN R Repositary* 2021.

2. Greenwell B, Boehmke B, Cunningham J, GBM D. GBM: Generalized Boosted Regression Models; R Package Version 2.1. 8; 2020. 2021.

3. Hastie T, Tibshirani R, Friedman JH, Friedman JH. The elements of statistical learning: data mining, inference, and prediction: Springer; 2009.

4. Majithia S, Tham Y-C, Chee M-L, et al. Cohort profile: the Singapore epidemiology of eye diseases study (seed). *International journal of epidemiology* 2021; 50(1): 41-52.

5. Levey AS, Coresh J, Balk E, et al. National Kidney Foundation Practice Guidelines for Chronic Kidney Disease: Evaluation, Classification, and Stratification. *Annals of Internal Medicine* 2003; 139(2): 137-47.

6. Reutens AT. Epidemiology of Diabetic Kidney Disease. *Medical Clinics* 2013; 97(1): 1-18.

7. Lim CC, He F, Li J, et al. Application of machine learning techniques to understand ethnic differences and risk factors for incident chronic kidney disease in Asians. *BMJ Open Diabetes Research and Care* 2021; 9(2): e002364.

8. Levey AS, Stevens LA, Schmid CH, et al. A new equation to estimate glomerular filtration rate. *Annals of internal medicine* 2009; 150(9): 604-12.

9. Soininen P, Kangas AJ, Würtz P, Suna T, Ala-Korpela M. Quantitative serum nuclear magnetic resonance metabolomics in cardiovascular epidemiology and genetics. *Circulation: cardiovascular genetics* 2015; 8(1): 192-206.

10. Würtz P, Kangas AJ, Soininen P, Lawlor DA, Davey Smith G, Ala-Korpela M. Quantitative serum nuclear magnetic resonance metabolomics in large-scale epidemiology: a primer on-omic technologies. *American journal of epidemiology* 2017; 186(9): 1084-96.

11. Sabanayagam C, Lim SC, Wong TY, Lee J, Shankar A, Tai ES. Ethnic disparities in prevalence and impact of risk factors of chronic kidney disease. *Nephrology Dialysis Transplantation* 2010; 25(8): 2564-70.

12. Sabanayagam C, Teo BW, Tai ES, Jafar TH, Wong TY. Ethnic variation in the impact of metabolic syndrome components and chronic kidney disease. *Maturitas* 2013; 74(4): 369-74.

13. Lim CC, He F, Li J, et al. Application of machine learning techniques to understand ethnic differences and risk factors for incident chronic kidney disease in Asians. *BMJ Open Diabetes Research and Care* 2021; 9(2): e002364.

14. Sabanayagam C, Shankar A, Saw SM, Lim SC, Tai ES, Wong TY. Socioeconomic status and microalbuminuria in an Asian population. *Nephrology Dialysis Transplantation* 2009; 24(1): 123-9.

15. Sabanayagam C, Teo BW, Tai ES, Jafar TH, Wong TY. Ethnic differences in the association between blood pressure components and chronic kidney disease in middle aged and older Asian adults. *BMC nephrology* 2013; 14: 1-11.

16. Sabanayagam C, Liew G, Tai E, et al. Relationship between glycated haemoglobin and microvascular complications: is there a natural cut-off point for the diagnosis of diabetes? *Diabetologia* 2009; 52: 1279-89.

17. Betzler BK, Sultana R, Banu R, et al. Association between body mass index and chronic kidney disease in Asian populations: a participant-level meta-analysis. *Maturitas* 2021; 154: 46-54.

18. Yip W, Ong PG, Teo BW, et al. Retinal vascular imaging markers and incident chronic kidney disease: a prospective cohort study. *Scientific Reports* 2017; 7(1): 1-9.

19. Sabanayagam C, Shankar A, Koh D, et al. Retinal microvascular caliber and chronic kidney disease in an Asian population. *American journal of epidemiology* 2009; 169(5): 625-32.

20. Sabanayagam C, Foo VHX, Ikram MK, et al. Is chronic kidney disease associated with diabetic retinopathy in A sian adults? *Journal of diabetes* 2014; 6(6): 556-63.

[Editors’ note: what follows is the authors’ response to the second round of review.]The manuscript has been improved but there are some remaining issues that need to be addressed, as outlined below:Reviewer #2 (Recommendations for the authors):The authors have answered most of the questions adequately. One major concern I still have is its longitudinal nature with 6 yr timeframe for the development of DKD. The authors have noted that as the AUC of the model is 0.85, baseline characteristics are good enough to predict DKD. If we look closely though the incidence of DKD was <12%. It is thus an unbalanced dataset and therefore AUC is likely an inflated value that needs to be interpreted with caution. I realize that ultimately this is the data that the authors have and it is still a valuable addition to the literature but this is an important limitation that needs to be acknowledged upfront.

We thank the Reviewer for his insightful comment. We have now included the following sentences as an additional limitation in page 18, paragraph 1 as follows:

“Fourth, the 6-year incidence of DKD in our cohort was relatively low, with less than 12% of participants developing DKD within that timeframe. It is thus possible that the unbalanced distribution of outcomes in our dataset may have inflated the value of AUC that needs to be interpreted with caution.”